# Transgelin-2 in Multiple Myeloma: A New Marker of Renal Impairment?

**DOI:** 10.3390/molecules27010079

**Published:** 2021-12-23

**Authors:** Karolina Woziwodzka, Jolanta Małyszko, Ewa Koc-Żórawska, Marcin Żórawski, Paulina Dumnicka, Artur Jurczyszyn, Krzysztof Batko, Paulina Mazur, Małgorzata Banaszkiewicz, Marcin Krzanowski, Paulina Gołasa, Jacek A. Małyszko, Ryszard Drożdż, Katarzyna Krzanowska

**Affiliations:** 1Chair and Department of Nephrology, Jagiellonian University Medical College, 30-688 Kraków, Poland; woziwodzka.karolina@gmail.com (K.W.); batko.krzysztof@gmail.com (K.B.); mbanaszkiewicz92@gmail.com (M.B.); mkrzanowski@op.pl (M.K.); paulinagolasa1@gmail.com (P.G.); 2Department of Nephrology, Dialysis and Internal Medicine, Medical University of Warsaw, 02-091 Warsaw, Poland; jolmal@poczta.onet.pl; 3Second Department of Nephrology and Hypertension with Dialysis Unit, Medical University of Bialystok, 15-276 Bialystok, Poland; ewakoczorawska@wp.pl; 4Department of Clinical Medicine, Medical University of Bialystok, 15-254 Bialystok, Poland; mzorawski@wp.pl; 5Department of Medical Diagnostics, Jagiellonian University Medical College, 30-688 Kraków, Poland; paulina.dumnicka@uj.edu.pl (P.D.); paulina.pater@uj.edu.pl (P.M.); ryszard.drozdz@uj.edu.pl (R.D.); 6Chair and Department of Hematology, Jagiellonian University Medical College, 31-501 Kraków, Poland; mmjurczy@cyf-kr.edu.pl; 7First Department of Nephrology and Transplantology with Dialysis Unit, Medical University of Bialystok, 15-540 Bialystok, Poland; jacek.malyszko@umb.edu.pl

**Keywords:** multiple myeloma, transgelin, tubular kidney injury, biomarker

## Abstract

Transgelin is a 22-kDa protein involved in cytoskeletal organization and expressed in smooth muscle tissue. According to animal studies, it is a potential mediator of kidney injury and fibrosis, and moreover, its role in tumorigenesis is emerging in a variety of cancers. The study included 126 ambulatory patients with multiple myeloma (MM). Serum transgelin-2 concentrations were measured by enzyme-linked immunoassay. We evaluated associations between baseline transgelin and kidney function (serum creatinine, estimated glomerular filtration rate—eGFR, urinary markers of tubular injury: cystatin-C, neutrophil gelatinase associated lipocalin—NGAL monomer, cell cycle arrest biomarkers IGFBP-7 and TIMP-2) and markers of MM burden. Baseline serum transgelin was also evaluated as a predictor of kidney function after a follow-up of 27 months from the start of the study. Significant correlations were detected between serum transgelin-2 and serum creatinine (R = 0.29; *p* = 0.001) and eGFR (R = −0.25; *p* = 0.007). Transgelin significantly correlated with serum free light chains lambda (R = 0.18; *p* = 0.047) and serum periostin (R = −0.22; *p* = 0.013), after exclusion of smoldering MM patients. Patients with decreasing eGFR had higher transgelin levels (median 106.6 versus 83.9 ng/mL), although the difference was marginally significant (*p* = 0.05). However, baseline transgelin positively correlated with serum creatinine after the follow-up period (R = 0.37; *p* < 0.001) and negatively correlated with eGFR after the follow-up period (R = −0.33; *p* < 0.001). Moreover, higher baseline serum transgelin (beta = −0.11 ± 0.05; *p* = 0.032) significantly predicted lower eGFR values after the follow-up period, irrespective of baseline eGFR and follow-up duration. Our study shows for the first time that elevated serum transgelin is negatively associated with glomerular filtration in MM and predicts a decline in renal function over long-term follow-up.

## 1. Introduction

According to the Global Burden of Disease 2016 study, the worldwide age-standardized incidence and death rate of multiple myeloma (MM) are estimated at 2.1 and 1.5 per 100,000 individuals, respectively. The highest incidence is observed in Australasia, North America and Western Europe, while between 1990 and 2016, the numbers of new cases and deaths were increasing [1]. MM is a proliferative plasma cell disorder that is more prevalent in the aging population and presents itself with characteristic features of organ involvement: bone lesions, anemia, renal insufficiency, hypercalcemia and specific malignancy biomarkers (plasma cell clonality ≥60%, involved to uninvolved serum free light chains (FLCs) ≥100 and >1 focal lesion on magnetic resonance imaging) [2]. Evidence of clonal plasma cell percentage of ≥10% in bone marrow or biopsy-proven plasmacytoma is necessary to define a case of MM [2]. Kidney involvement in MM is common (close to one fourth of patients) and can carry a poor prognosis, particularly if renal function does not recover [3,4,5]. Irreversible renal failure is less frequent, but may be present in up to 8% of patients [3]. Severe renal failure is a deleterious condition with a high risk of early death and is one of the major culprits of early mortality [6,7]. A low treatment response rate and a median survival of 3–4 months was observed prior to the emergence of modern myeloma therapies [8,9,10]. Chemotherapy response and severity of renal failure are independent predictors of survival, which highlights the importance of early and efficacious treatment targeting the plasma cell clone to prevent renal damage [3]. Reversibility of renal failure may be as significant as the response to chemotherapy in terms of prognosis [9], although recent studies have shown that even if reversal is achieved, the outcomes are inferior to patients with normal renal function at baseline [11]. Renal insufficiency is considered reversible in about 50% of cases in some reports [8,9]. With the advent of novel drugs (e.g., proteasome inhibitors), prognosis for MM patients with renal insufficiency has remarkably improved [7,11,12,13,14,15,16].

In clinical routine, the laboratory assessment of renal function usually largely relies on serum creatinine measurements used to estimate the glomerular filtration rates. However, routinely available diagnostic measures (e.g., serum creatinine) are subject to several caveats and do not always allow for early and reliable prediction of ongoing renal damage [17]. Serum creatinine concentrations should be interpreted with the awareness of confounding factors such as (1) prerenal azotemia, (2) production rate dependent on individual and clinical characteristics, i.e., age, gender, muscle mass, and medication use, (3) late rise in up to 72 h following injury, (4) large “renal reserve”, and (5) concomitant diseases (e.g., sepsis, liver disease, muscle wasting). Therefore, serum creatinine may not adequately reflect the actual fall in glomerular filtration. Novel biomarkers that will aid clinicians in predicting renal injury and chronic kidney disease development are of high interest. Research to identify renal injury (RI) mediators or markers in the specific context of myeloma is warranted as it may also aid efforts to create models, including different pathophysiological processes leading to RI in MM patients (i.e., tubulitis vs glomerulonephritis).

Transgelin-2 (SM22), a cytoskeletal actin-binding protein involved in differentiation of smooth muscle cells, osteoblasts and adipocytes, is present in fibroblasts, some epithelial cells, immune cells (as the only one from transgelin family proteins), bone marrow cells or stem cells [18]. Basic function of transgelin-2 is participation in cytoskeleton remodeling via its effect on actin regulation. Moreover, transgelin is involved in bone marrow mesenchymal stem cell (MSC) proliferation and differentiation. Recent studies report dysregulation of SM22 in different malignancies and emphasize its role in cancer development and progression. According to available data, apart from its oncogenic role in solid tumors, SM22 is also upregulated in leukemia and lymphoma cell lines and takes part in B-cell lymphoma development [19]. Interestingly, transgelin-2 overexpression may be associated with chemotherapy resistance [20].

Furthermore, transgelin-2 was found to be a marker of interstitial fibrosis, glomerulosclerosis and renal damage [21]. Its upregulation depends on the etiology of the disease in various cells (glomerular parietal or visceral, or tubular interstitial cells) and elevated expression was detected in both glomerular and tubulointerstitial injury.

The aim of our study was to evaluate serum transgelin as a potential marker of renal impairment in patients with MM. We hypothesized that increased serum concentrations of transgelin may be related to irreversible renal insufficiency in this disease.

## 2. Results

### 2.1. Characteristics of the Studied Group

Our prospective observational study included 126 patients with MM, recruited during ambulatory control visits at the Departments of Hematology of the University Hospital in Kraków, Poland. A total of 73 women and 53 men were included, aged 29 to 90 years (Table 1). Symptomatic MM was diagnosed in 119 patients, the majority of whom were in ISS stage I. The remaining seven patients had smoldering MM. Most patients underwent at least one line of chemotherapy before the start of the study and roughly half of the group received maintenance treatment at baseline (Table 1). A history of acute kidney injury was positive in eight (6%) patients.

Mean baseline eGFR (CKD-EPI_Cr_) was 74 (SD: 24) mL/min/1.73 m^2^ and 29 (23%) patients had eGFR <60 mL/min/1.73 m^2^ (Table 2). eGFR was above 60 mL/min/1.73 m^2^ in 97 patients (77%), between 30 and 60 mL/min/1.73 m^2^ in 18 patients (14%) and below 30 mL/min/1.73 m^2^ in 11 patients (9%). The selected laboratory test results in the studied group are shown in Table 2. As compared to a control group of 32 healthy volunteers (16 men and 16 women aged 29–74 years), the studied MM patients presented with significantly higher serum concentrations of transgelin and interleukin 6, and higher urinary concentrations of IGFBP-7 and TIMP-2 (Table 2). Although the age range of healthy controls was matched with the age range of patients, the mean age was lower in the control group (50 versus 67 years, *p* < 0.001). However, serum transgelin remained higher in patients then in controls after adjustment for the age difference (*p* = 0.034). The sex distribution in the study and control groups was similar (*p* = 0.4).

### 2.2. Variables Associated with Serum Transgelin at Baseline

Median serum transgelin in the whole studied group was 84.1 ng/mL (Table 2). Transgelin concentrations were higher in men (median 96.2 versus 78.8 ng/mL; Figure 1A), in patients with smoldering MM (median 149.2 versus 82.4 ng/mL; *p* = 0.003; Figure 1B) and in treatment-naïve patients (median 145.2 versus 82.5 ng/mL; *p* = 0.014; Figure 1C); however, the majority of patients with smoldering MM (5 out of 7) and treatment-naïve patients (6 out of 8) were men. Of interest, serum transgelin was higher in healthy women than in men (control group; median 83.4 vs 60.5 ng/mL, respectively; *p* = 0.011).

For patients who had received any MM treatment before entering the study, there was no correlation between serum transgelin and the number of prior treatment lines (R = −0.10; *p* = 0.3). Patients on the maintenance treatment had non-significantly lower serum transgelin than those treatment-naïve (median 77.4 versus 91.0 ng/mL; *p* = 0.051). Treatment regimens included lenalidomide in 25 patients (20%), bortezomib in 21 (17%), thalidomide in 15 (12%), cyclophosphamide in 8 (6%), and melphalan in 3 (2%); steroids were used in 58 (46%) of patients. We did not observe any significant differences in serum transgelin concentrations between patients treated versus untreated with particular drugs (*p* > 0.05 for all comparisons). Transgelin levels did not differ according to ISS stage (*p* = 0.3; Figure 1B) or disease status (complete or partial response, stable disease or progressive disease; *p* = 0.2). Serum transgelin in patients with eGFR <60 mL/min/1.73 m^2^ did not differ significantly from the levels observed in those with higher eGFR (median 106.6 versus 79.7 ng/mL; *p* = 0.057; Figure 1D). The presence of proteinuria was not associated with serum transgelin (*p* = 0.3).

Significant correlations were detected between transgelin and serum creatinine (R = 0.29; *p* = 0.001), eGFR (CKD-EPI_Cr_) (R = −0.25; *p* = 0.007), uric acid (R = 0.19; *p* = 0.036), alanine (R = 0.18; *p* = 0.048) and aspartate (R = 0.26; *p* = 0.003) aminotransferases, ferritin (R = −0.22; *p* = 0.049), hepcidin (R = −0.25; *p* = 0.033), and urine cystatin C (R = 0.19; *p* = 0.042). Moreover, after exclusion of patients with smoldering MM, transgelin significantly correlated with serum FLC lambda (R = 0.18; *p* = 0.047) and serum periostin (R = −0.22; *p* = 0.013). In multiple forward stepwise regression, uric acid was identified as the only independent predictor of serum transgelin (beta = 0.31; standard error = 0.13; *p* = 0.023).

### 2.3. The Association between Serum Transgelin and Renal Function after the Follow-Up Period

The follow-up data were collected for 27 months from the start of the study. A total of 23 patients (18%) died during the study period, including 12 due to MM, 7 due to infection, and 3 due to undefined reasons. The median observation time was 21 (15; 24) months. At the end of the follow-up period, eGFR was lower in 47 (37%) of patients and increased or remained unchanged in 71 (56%) patients. No follow-up data were available for eight patients (6%). The patients in whom eGFR decreased tended to present higher baseline transgelin as compared to those with no change or an increase in eGFR values (median 106.6 versus 83.9 ng/mL; *p* = 0.051; Figure 2).

However, baseline transgelin positively correlated with serum creatinine after follow-up (R = 0.37; *p* < 0.001) and negatively correlated with eGFR after follow-up (R = −0.33; *p* < 0.001; Figure 2). Moreover, higher baseline serum transgelin significantly predicted lower eGFR values after the follow-up period, independently of baseline eGFR, urinary concentrations of tubular injury markers (NGAL monomer and IGFBP-7), sex, age, prior treatment, treatment response and observation duration (Table 3). Moreover, transgelin values in the upper tertile (i.e., above 110.6 ng/mL) were independently associated with lower eGFR at the end of observation (Figure 2; Table 3). Although baseline uNGAL (R = −0.31; *p* < 0.001) and uIGFBP-7 (R = −0.35; *p* < 0.001) were also significantly associated with final eGFR, these associations did not prove to be independent of other predictors, including baseline eGFR (Table 3).

By using simple Cox regression, we did not observe any association between serum transgelin and overall- or MM-specific survival, neither in the entire study group, nor after exclusion of patients with smoldering MM (*p* > 0.5 in all cases).

## 3. Discussion

Recent findings suggest that transgelin may be a potential player in fibrosis and a marker of kidney injury. It has been investigated in various kidney diseases [22,23,24,25,26,27]. Experimental data in animal models of anti-glomerular basement membrane nephritis revealed that SM22α expression may reflect structural and functional shifts following injury. Downregulation of particular podocyte proteins and expression of transgelin may reflect dedifferentiation and transdifferentiation of the injured glomerular epithelium [24,26]. In chronic renal injury models (5/6 nephrectomy) with early tubulointerstitial injury, SM22α expression is observed early in the peritubular and periglomerular compartments. In the ischemia-reperfusion setting, which mainly affects the tubular epithelium, SM22α expression was noted in the peritubular interstitium [24]. In obstructive nephropathy models, periglomerular fibroblasts were observed as the primary cells with transgelin up-regulation, with subsequent elevation in interstitial fibroblasts [28]. These data show transgelin expression in both glomerular and tubulointerstitial injury, which is not limited to a single cell type and can be considered a general indicator of kidney insult. Taken together, these data suggest that in the chronic cycle of injury, repair and scarring of kidney tissue, transgelin may be a marker that reflects this process.

The salient finding of this study is the relationship between the novel biomarker transgelin-2 and progression of renal impairment over a median 21-month observation in patients with MM. Higher baseline serum transgelin predicted lower eGFR by the end of the follow-up, irrespective of baseline eGFR, urinary concentrations of tubular injury markers (NGAL monomer and IGFBP-7), sex, age, prior treatment, treatment response and observation duration. To our best knowledge, this is the first study evaluating the potential utility of this new molecule related to kidney injury and its associations with established indices of renal function in patients with MM at different stages of management. Previous studies focused on transgelin-2 oncogenic potential and associations with MM transformation to plasma cell leukemia (PCL) that emphasize transgelin-2 role as a poor survival marker [19]. However, we were not able to prove the association between serum transgelin concentrations and survival.

At the start of the study, serum transgelin correlated positively with serum creatinine and negatively with eGFR. Although we have measured urinary concentrations of several markers of tubular injury, i.e., NGAL monomer, cystatin C and cell cycle arrest biomarkers: TIMP-2 and IGFBP-7, we only found weak significant correlation between serum transgelin and urine cystatin C. Moreover, serum transgelin correlated positively with serum concentrations of FLC lambda (the type of FLC more often associated with renal injury in MM). Case studies on kidney biopsies in MM indicate a heterogenous spectrum of kidney lesions, with myeloma cast nephropathy (MCN) as the most common condition [17,18]. Experimental evidence suggests that FLCs play a crucial role in the induction of pro-inflammatory and fibrotic changes within the kidney compartment [29,30]. High levels of serum FLCs underlie this MM-specific lesion, which has translated into clinical utility of reducing FLCs and the corresponding renal recovery [31]. Data indicate that specific renal biopsy findings are also associated with prognosis in MM [32,33]. Cast formation and interstitial fibrosis, as well as tubular atrophy (IFTA), have been adversely related to renal recovery in multivariate models including hematological status and clinical characteristics [33]. Histopathology characteristics (including IFTA) have been previously studied along with clinical variables in models predicting kidney failure [34]. It has been emphasized that initial clinical appraisal does not correspond well to the underlying pathology, which highlights the importance of kidney biopsy and reliable biomarkers that could help to localize (e.g., proximal tubule injury, tubulointerstitial injury) and define the lesion. Identifying novel markers that could reflect developing nephropathology is of high interest, as it can facilitate early diagnosis and may guide treatment choice. Ideally, development of non-invasive instruments, specific for distinct kidney lesions, could support treatment decisions and risk stratification in the future.

The diagnosis of MM is based on hematological analysis, mainly bone marrow biopsy. In clinical practice, kidney biopsy is not a mandatory procedure for selection of the appropriate treatment regimen. Taking into consideration the invasiveness of kidney biopsy, potential kidney injury markers may be a favorable solution for patients with MM. However, our main finding was the association between baseline serum transgelin and final eGFR, which shows that circulating transgelin concentrations predict long-term irreversible kidney insufficiency in patients with MM.

In our patients with MM, serum transgelin was significantly higher, as compared to healthy controls. We also observed higher transgelin levels in men with MM, as compared to women. The sex-related difference could not be attributed to differences in MM stage, kidney function or treatment as these were not different between men and women (data not shown). Sex-related differences in circulating transgelin, however, have been reported by others. The proteomic analysis of human plasma published by Silliman at al. [35] revealed 14-fold higher transgelin concentrations in males than in females. Animal studies also reported sex-related differences in the expression of transgelin [36]. Interestingly, in our control group trangelin levels were higher in females. We do not know the reasons for sex-related differences in circulating transgelin concentrations, and further studies are necessary to reveal the underlying causes.

Transgelin is involved in cytoskeletal organization and contractility [37], and is expressed in smooth muscle cells [38,39]. This actin-binding protein is also under scrutiny with regard to cancer cell proliferation, invasion and metastases. Actin is a central element of the cytoskeleton that is involved in a variety of functions, while its disorganization and rearrangement is involved in cancer pathology. There are three types of transgelin proteins (type 1, 2 and 3). Type 2 is abundant in smooth muscle cells and was initially described as SM22α. Transgelin-2 expression is up-regulated in several cancers with its staining being higher in tumor cells rather than in tumor stroma. Moreover, transgelin-2 is involved in bone marrow mesenchymal stem cell (MSC) proliferation and differentiation. Several studies revealed transgelin upregulation in leukemia and lymphoma cell lines. Although transgelin-2 overexpression has been associated with chemotherapy resistance, the precise mechanism is not known. According to previous studies, overexpressed trangelin-2 gene was found in methotrexate-resistant human choriocarcinoma cells and paclitaxel-resistant human breast cancer cells [20]. Transgelin-2 overexpression has also been linked with poor prognosis, and transgelin-2 has been proposed as a potential treatment target due to its restriction to tumor cells (to the contrary to transgelin type-1) [40].

The most popular three-drug regimen used in the MM treatment consists of: a proteasome inhibitor-bortezomib, and immunomodulatory drugs-lenalidomide and dexamethasone. In case of relapse or refractory MM, antibodies targeting myeloma cells (e.g., daratumumab, elotuzumab, isatuximab, belantamab mafodotin), nuclear export inhibitors (selinexor) or histone deacetylase inhibitors (panobinostat) are used. One of the most important game-changer drugs in MM was proteasome inhibitor bortezomib, owing to its various anti-myeloma effects including disruption of the cell cycle and induction of apoptosis, alteration of the bone marrow microenvironment and inhibition of nuclear factor kappa B (NFκB). This novel agent improves renal function and should be used especially in the group of patients with lower GFRs [41]. Moreover, Bolomsky et al. found an association between gene expression levels of several immunomodulatory drug targets in bone marrow mononuclear cells of MM patients and a response to the lenalidomide-dexamethasone regimen [42]. Interestingly, high IKAROS protein levels are associated with successful outcome in MM patients [42]. IKAROS was also found among smooth muscle genes in renin cells in the kidney [43].

In our study, patients were treated with various regimens, most commonly including lenalidomide (in 20% of patients). Serum concentrations of transgelin did not differ according to drugs used. Although we observed higher concentrations of transgelin in patients who did not receive any MM treatment before the study, in those previously treated there was no association between serum transgelin and the number of treatment lines received. Specifically, serum transgelin levels did not differ between those who received lenalidomide and those who did not. Lenalidomide modulates different components of the immune system by interactions with cytokine production through T-cell and NK cell regulation. It is associated with inhibition of pro-inflamatory cytokines interleukin 6 and tumor necrosis factor α (TNF-α). Furthermore, lenalidomide inhibits MM cells and their interactions, leading to apoptosis [44]. In our study, there was no correlation between serum interleukin 6 and transgelin levels in MM patients (R = 0.09; *p* = 0.4). Considering that only 25 patients received lenalidomide at the start of our study, we cannot reliably exclude a weak to moderate effect of the drug (or other anti-MM medications) on serum transgelin concentrations. Since transgelin expression has been studied only as an overexpressed molecule in MM transformation to PCL, future studies should reveal the role of transgelin in the development of MM and how it may be affected by MM treatment.

Although smoldering MM (SMM) patients have normal renal function defined as GFR >60 mL/min/1.73 m^2^, we found elevated trangelin-2 serum concentrations in this group. This finding may have a pathophysiological explanation. The upregulation of transgelin-2 has been associated with tumorigenesis and cancer development and may vary along with clinical stage and tumor size. Interestingly, several studies revealed higher levels of transgelin-2 in inflammation (i.e., SIRS) and explored SM22 overexpression in the regulation of the NIK transcription and proinflammatory NF-kB-signaling pathways as a modulator of vascular inflammation [20,45]. These studies suggest that transgelin may be viewed as an anti-inflammatory marker. Taking into consideration the role of interleukin 6 in MM pathogenesis as a growth and survival factor, inhibiting apoptosis in myeloma cells, this may also explain SM22 role in tumorigenesis. This may support the hypothesis that at the beginning of the disease and tumorigenesis, transgelin-2 concentrations are higher. However, a few reports demonstrated that transgelin-2 inhibits the motility of cancer cells by suppressing actin polymerization. Moreover, according to available data, only 2% of patients with SMM develop MM. Further, higher concentrations of transgelin in our patients with SMM may possibly be associated with the fact that they had received no treatment. Transgelin levels were also higher in patients who did not received any MM treatment before the study. Moreover, the sex of the patients with SMM may play a role in elevated transgelin-2 concentrations as in the studied group transgelin concentrations were higher in men, and SMM/untreated patients were mostly men. However, since we were not able to identify previous reports on transgelin in patients with SMM, and the number of patients with SMM in our study is very low, we may only speculate on this finding.

There are several limitations of the present study that have to be emphasized. First, the study group is a heterogenous sample of patients from the outpatient clinic. Differences in individual and disease characteristics can mask the relationship of transgelin-2 with renal involvement. Moreover, there are no standardized laboratory assays to measure transgelin concentrations. Although we provided the information on serum transgelin in a small group of healthy persons, these data must be considered provisional. Transgelin-2 has been scarcely studied in MM and understanding of its mechanistic role and potential place as a marker of kidney injury requires differentiation of the major source of this molecule in circulation. Subsequent studies should investigate kidney biopsy samples to reveal the association between transgelin and various types of kidney injury in MM, and should validate our findings in a comparative study with renal involvement and control population. 

## 4. Materials and Methods

### 4.1. Study Design and Patients

This was a prospective observational study. Patients were recruited during ambulatory control visits at the Departments of Hematology of the University Hospital in Kraków, Poland. Age ≥ 18 years and the diagnosis of SMM or MM according to International Myeloma Working Group represented the inclusion criteria. The exclusion criteria were the following: (1) recent active infection, (2) a history of hepatitis B, C, or human immunodeficiency virus (HIV) infection, and (3) cancers other than myeloma. For all patients, detailed history of the disease was collected from available medical records. Data collected at the initial study visit included patient’s age and sex, the date of initial diagnosis of SMM or MM, the current diagnosis, the presence of bone lesions on X-ray, and the information about past and present treatment including the response to the treatment: complete response (CR), partial response (PR), stable disease (SD), or progressive disease (PD). The follow-up data were collected after 27 months from the start of the study, and included (1) the date and the cause of death, (2) the results of laboratory tests, including serum creatinine and eGFR obtained at the final follow-up visit.

Control group was recruited in order to obtain the reference results of non-standard laboratory tests and included 32 healthy volunteers (16 women, 16 men) aged 29 to 74 years.

### 4.2. Ethics Statement

The study was conducted according to the principles of the Declaration of Helsinki and in compliance with the International Conference on Harmonization/Good Clinical Practice regulations. The study was approved by the Bioethics Committee of the Jagiellonian University and all patients signed an informed consent for their participation. All patients were treated at the Department of Hematology, University Hospital in Krakow.

### 4.3. Blood Samples and Laboratory Tests

In the morning of the day of blood collection following overnight fasting and rest, routine laboratory tests of both patients and control subjects were performed, which included complete blood counts, serum concentrations of creatinine, serum activity of lactate dehydrogenase, total protein, albumin, β2-microglobulin, free light chains, urine concentrations of light chains, alanine and aspartate aminotransferases, and ferritin.

Blood was collected based on Good Laboratory Practice (GLP) and Good Clinical Practice (GCP) principles by qualified staff. Blood was collected to closed tubes with clot activator. The samples were mixed and kept in the ambient temperature in vertical position for 30 min, then centrifuged at 1300× *g* for 15 min. After centrifugation, serum was aliquoted and kept at −70 °C until analysis. We did not use hemolyzed or lipemic samples.

Serum samples for other laboratory tests were aliquoted and stored at a temperature below −70°C. These non-routine laboratory tests included urine NGAL monomer, urine IGFBP-7 and TIMP-2, urine cystatin-C, transgelin, periostin, hepcidin and interleukin 6 (IL-6).

Automatic biochemical analysers, Hitachi 917 (Hitachi, Ibaraki, Japan) and Modular P (Roche Diagnostics, Mannheim, Germany), were used. Hematological parameters were measured using a Sysmex XE 2100 analyser (Sysmex, Kobe, Japan). The concentration of serum FLC, urine LC (κ and λ type) and β2-microglobulin were measured by immunonephelometric method on a BN II analyser (Siemens GmbH, Goerlitz, Germany). The determination of free light chains (FLC κ, FLC λ) were performed using Freelite reagents (Binding Site, Birmingham, UK) with reference ranges: 1.7–3.7 g/L and 0.9–2.1 g/L respectively. The immunophenotype of monoclonal protein was determined by serum immunofixation (IFE) on agarose gel (EasyFix G26, Interlab, Italy).

The eGFR value was calculated based on serum creatinine using Chronic Kidney Disease-Epidemiology Collaboration (CKD-EPI) 2009 formula based on serum creatinine and cystatin C.

The non-routine laboratory tests were performed in series, using commercially available immunoenzymatic test kits. Serum IL-6 was measured using Quantikine ELISA Human IL-6 Immunoassay (R&D Systems, Inc., Minneapolis, MN, USA), with the minimum detectable level of 0.70 pg/mL, and the intra- and interassay precision of 2.0% and 3.8%, respectively. The reference range for IL-6 was 3.13–12,5 pg/mL. Serum transgelin was measured using Human TAGLN (Transgelin) ELISA Kit (Wuhan Fine Biotech Co, Wuhan, China), with the detection range of 0.625–40 ng/mL, and the intra- and interassay precision of 8% and 10%, respectively, as reported by the manufacturer of the assay. Serum periostin was measured using Periostin Enzyme Immunoassay Kit (Biomedica Medizinprodukte GmbH &Co KG, Wien, Austria), with the detection range of 0–4000 pmol/L; limit of detection of 20 pmol/L and the intra- and interassay precision of 3.0% and 6.0% respectively. Serum hepcidin 25 levels were measured using Hepcidin 25 human Cet. No. S-1337 kit (Peninsula Laboratories International, Inc., San Carlos, CA, USA). The detection range for hepcidin 25 is 0.02–25 ng/mL. Urine NGAL monomer was assessed using Human NGAL monomer-specific ELISA Kit (BioPorto Diagnostics A/S, Hellerup, Denmark), with the minimum detectable dose of 10 pg/mL and the detection range for NGAL of 10–1000 pg/mL. Urine IGFBP-7 was measured using IBP-7 ELISA Kit (EIAab Science Inc, Wuhan, China). The detection range for IGFBP7 was 0.312–20 ng/mL. Urine TIMP-2 levels were measured using Human Metalloproteinase inhibitor 2 ELISA Kit (EIAab Science Inc, Wuhan, China), with the detection range of 0.312–20 ng/mL. Cystatin C concentrations in urine and serum were measured using Human Cystatin C ELISA- IVD (BioVendor Research and Diagnostic Products, Brno, Czech Republic), with the detection range of 0.25–25 ng/mL, and the intra- and interassay precision of 3.5% and 10.4%, respectively.

### 4.4. Statistical Analysis

Data were reported as number of patients and percentages of the group for categories and mean (standard deviation, SD) or median (lower; upper quartile) for quantitative variables with or without normal distribution, respectively. The Shapiro-Wilk’s test was used to assess normality. As serum transgelin values were not normally distributed, they were compared between groups with Mann-Whitney’s or Kruskal-Wallis’ test. Age-adjusted logistic regression was used to verify serum transgelin level difference between patients and controls. Spearman rank correlation coefficient was used to assess simple correlations. Stepwise backward multiple linear regression was used to look for independent predictors of serum transgelin level. The model included the variables significantly (*p* < 0.05) correlated with serum transgelin in simple analysis. Multiple linear regression was also used to evaluate the association between baseline serum transgelin and eGFR values at the end of the follow-up. Serum transgelin was included in the models either as a continuous variable or after categorization into tertiles. The models were adjusted for the variables significantly correlated with final eGFR values in simple analysis and pre-specified clinically relevant confounders (age, sex, previous treatment, treatment response, observation duration). Right-skewed variables were log-transformed before inclusion into linear regression models. Simple proportional hazard Cox regression was used to check for the association between serum transgelin concentrations and overall and MM-specific survival. The survival time was calculated from the start of the study until patient’s death or the date of his/her last follow-up. The statistical tests were two-tailed and *p* < 0.05 indicated statistical significance. Statistica 12.0 (StatSoft, Tulsa, OK, USA) was used for computations.

## 5. Conclusions

Our study shows for the first time that elevated serum transgelin is negatively associated with glomerular filtration rate in MM and predicts a decrease in renal function over long-term follow-up. However, elevated serum transgelin in MM might be associated with other pathological processes, e.g., malignancy, or inflammation, as well as with MM treatment. Considering the limitations of our study, further work is needed to evaluate transgelin expression in various types of renal involvement in MM. Still, our findings support previous reports that link transgelin to kidney fibrosis.

## Figures and Tables

**Figure 1 molecules-27-00079-f001:**
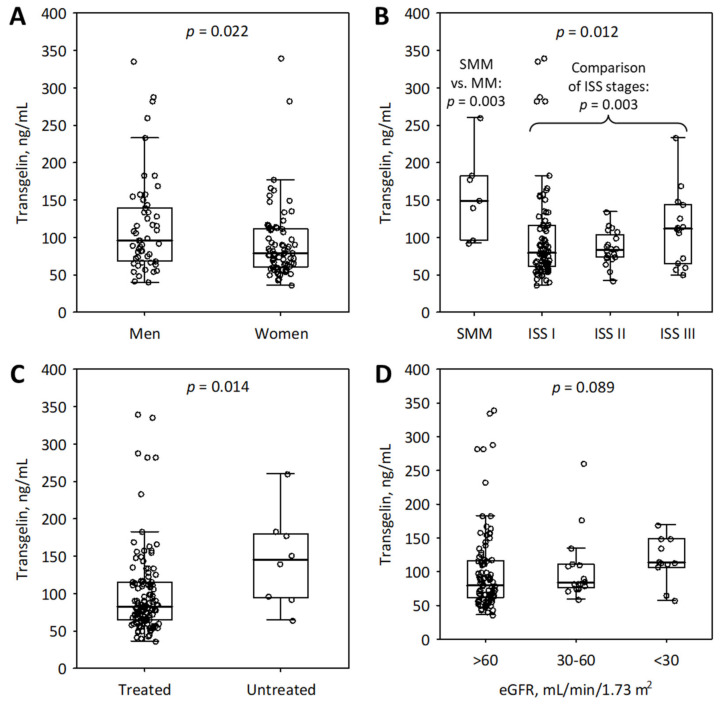
Associations between serum transgelin and sex (**A**), MM stage (**B**), previous treatment (**C**), and eGFR category (**D**) among the studied group of 126 patients with MM. Data are shown as median (central line), interquartile range (box), non-outlier range (whiskers); circles represent raw data. ISS, International Staging System; eGFR, estimated glomerular filtration rate; MM, multiple myeloma; SMM, smoldering MM.

**Figure 2 molecules-27-00079-f002:**
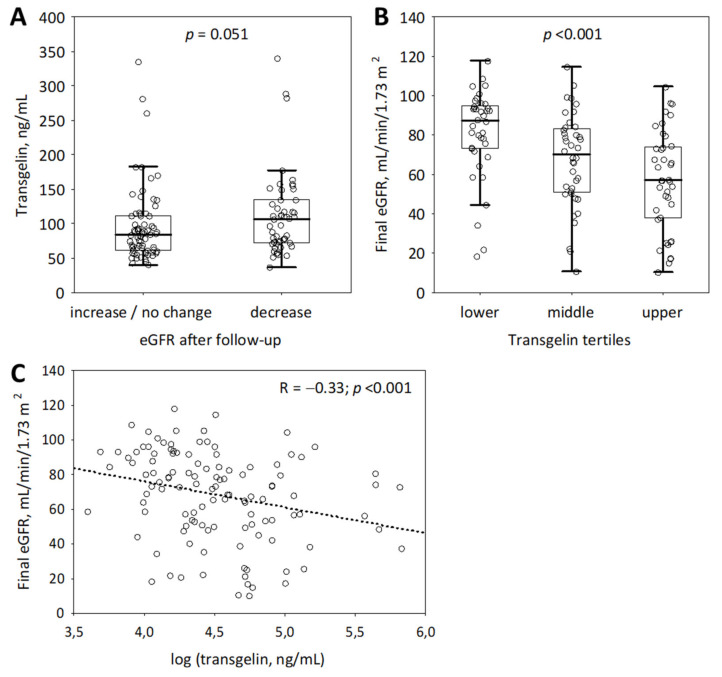
Associations between baseline serum transgelin and final eGFR (CKD-EPI_Cr_) values in the group of 126 patients with MM: baseline transgelin by eGFR changes (no change or decrease versus increase during the observation period; (**A**); final eGFR values by transgelin tertiles (the tertiles’ boundaries were 71.0 and 110.6 ng/mL; (**B**) and the correlation between baseline transgelin and final eGFR (**C**). Data on panels (**A**,**B**) are shown as median (central line), interquartile range (box), non-outlier range (whiskers); circles represent raw data. CKD-EPI_Cr_, Chronic Kidney Disease Epidemiology Collaboration equation based on serum creatinine; eGFR, estimated glomerular filtration rate; MM, multiple myeloma.

**Table 1 molecules-27-00079-t001:** Baseline clinical characteristics of the study group.

Characteristic	Results in the Studied MM Patients (*n* = 126)
Mean age (standard deviation), years	67 (10)
Male sex, *n* (%)	53 (42)
Median time since diagnosis (Q1; Q3), months	30 (14; 63)
Smoldering MM, *n* (%)	7 (6)
ISS:	
stage I, *n* (%)	84 (67)
stage II, *n* (%)	20 (16)
stage III, *n* (%)	15 (12)
Immunofixation:	
IgG, *n* (%)	92 (73)
IgM, *n* (%)	2 (2)
IgA, *n* (%)	23 (18)
κ, *n* (%)	79 (63)
λ, *n* (%)	44 (35)
free light chains, *n* (%)	18 (14)
biclonal, *n* (%)	5 (4)
non-secretory, *n* (%)	5 (4)
Disease state:	
CR, *n* (%)	41 (33)
PR, *n* (%)	48 (38)
SD, *n* (%)	9 (7)
PD, *n* (%)	28 (22)
On maintenance treatment, *n* (%)	61 (48)
Prior treatment:	
none, *n* (%)	8 (6)
1 line, *n* (%)	35 (28)
2 or more lines, *n* (%)	83 (66)
History of auto-PBSCT, *n* (%)	57 (45)
Anemia, *n* (%)	25 (20)
Bone lesions, *n* (%)	77 (61)
History of AKI, *n* (%)	8 (6)

AKI, acute kidney injury; CR, complete response; Ig, immunoglobulin; ISS, International Staging System; MM, multiple myeloma; PBSCT, peripheral blood stem cell transplantation; PD, progressive disease; PR, partial response; SD, stable disease; Q1, lower quartile; Q3, upper quartile

**Table 2 molecules-27-00079-t002:** Baseline results of laboratory tests in the studied group in comparison to reference ranges used in the study center for routine laboratory tests or values observed in 32 healthy controls for non-standard laboratory tests (^c^). Data are shown as mean (standard deviation) or median (lower; upper quartile) unless otherwise stated.

Laboratory Test	Results in MM Patients (*n* = 126)	Reference Range or Results in Healthy Controls (*n* = 32)
Creatinine, µmol/L	78 (67; 98)	F: 44–80; M: 62–106
eGFR (CKD-EPI_Cr_), mL/min/1.73 m^2^	73 (62; 91)	≥90
eGFR (CKD-EPI_Cr_) <60 mL/min/1.73 m^2^, *n* (%)	29 (23)	-
eGFR (CKD-EPI_Cr-CysC_), mL/min/1.73 m^2^	76 (27)	≥90
Uric acid, µmol/L	289 (81)	F: 143–340; M: 202–416
Albumin, g/L	42.0 (4.7)	35–52
β2-microglobulin, mg/L	2.54 (2.10; 3.63)	1.09–2.53
Involved serum FLC, mg/L	29.6 (16.3; 100.0)	κ: 6.7–22.4; λ: 8.3–27.0
Involved urine LC, mg/L	6.8 (6.3; 30.0)	κ: <7.09; λ: <3.89
White blood cell count, ×10^3^/µL	6.12 (4.89; 7.25)	4.0–10.0
Hemoglobin, g/dL	12.6 (1.7)	F: 11–15; M: 12–17
Platelet count, ×10^3^/µL	172 (143; 213)	125–340
Lactate dehydrogenase, U/L	355 (303; 404)	240–480
Interleukin 6, pg/mL ^a^	2.97 (1.61; 6.00)	1.51 (1.07; 2.05) ^c,^*
Ferritin, µg/L ^b^	164 (63; 414)	13–400
Hepcidin, ng/mL ^a^	28.8 (16.5; 44.6)	27.1 (20.0; 37.2) ^c^
Alanine aminotransferase, U/L	21 (16; 29)	F: 5–33; M: 5–41
Aspartate aminotransferase, U/L	21 (17; 27)	F: 5–32; M: 5–40
Transgelin, ng/mL	84.1 (65.4; 116.4)	69.3 (56.8–90.4) ^c,^*
Periostin, pmol/L	1133 (798; 1663)	1176 (995–1455) ^c^
Proteinuria, *n* (%)	24 (19)	-
Urine IGFBP-7, ng/mL	5.19 (2.24; 12.74)	2.65 (1.36–5.75) ^c,^*
Urine TIMP-2, ng/mL	2.60 (0.48; 8.78)	1.08 (0.15; 2.35) ^c,^*
Urine cystatin C, ng/mL	42.6 (16.3; 86.5)	46.7 (26.5–64.3) ^c^
Urine NGAL monomer, ng/mL	9.23 (4.42; 26.8)	9.06 (4.73–11.86) ^c^

^a^ available in a subgroup of 73 patients; ^b^ available in a subgroup of 82 patients; ^c^ median (lower; upper quartile) in a control group of 32 healthy volunteers; * *p* < 0.05 for the comparison between MM patients and healthy controls CKD-EPI, Chronic Kidney Disease Epidemiology Collaboration equation based on serum creatinine (Cr) or serum creatinine and cystatin C (Cr-CysC); eGFR, estimated glomerular filtration rate; FLC, free immunoglobulin light chains; IGFBP-7, insulin-like growth factor-binding protein-7; LC, immunoglobulin light chains; NGAL, neutrophil gelatinase-associated lipocalin; TIMP-2, tissue inhibitor of metalloproteinase-2.

**Table 3 molecules-27-00079-t003:** Linear regression models to predict eGFR (CKD-EPI_Cr_) values at the end of observation (median follow-up of 21 months) among 126 patients with MM. In Model 1, serum transgelin was included as a continuous variable after log-transformation, while in Model 2, serum transgelin was included after categorization into tertiles (the tertiles’ boundaries were 71.0 and 110.6 ng/mL).

Independent Variable	Model 1		Model 2	
	Beta ± SE	*p*-Value	Beta ± SE	*p*-Value
Baseline eGFR	0.75 ± 0.07	<0.001	0.71 ± 0.07	<0.001
log (baseline transgelin)	−0.14 ± 0.05	0.011	not included	
Transgelin tertiles:	not included			
lower			reference	-
middle			−0.07 ± 0.06	0.3
upper			−0.20 ± 0.07	0.003
log (urine NGAL monomer)	−0.05 ± 0.06	0.4	−0.06 ± 0.06	0.4
log (urine IGFBP-7)	−0.01 ± 0.06	0.9	−0.01 ± 0.06	0.9
Adjusted R^2^ for the model	0.72	<0.001	0.72	<0.001

Both models were additionally adjusted for sex, age, prior treatment, treatment response (complete or partial response, stable disease or progressive disease) and observation duration. The results are reported as standardized correlation coefficient (beta) and standard error (SE).

## Data Availability

The data is available from the corresponding author upon reasonable request.

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
