# Peer review of "Transgelin-2 in Multiple Myeloma: A New Marker of Renal Impairment?"

_molecules, 2021, doi:10.3390/molecules27010079_

Round 1

Reviewer 1 Report

Abstract: needs some work, the methods and patient samples used need to be more clearly defined in this section

Introduction: What was the reason to investigate transgelin in this study? Was this based on a literature review or had this group some previous findings on transgelin that prompted this study?

Serum creatinine measurement is subject to several caveats to prediction of ongoing renal damage. These caveats need to be explained in detail to put into perspective the need for additional biomarkers of kidney damage.

Results: The patient group studied in this manuscript represent a heterogenous population, how this this affect the analysis and were any inter-group comparison investigated based on response rates?

Table 2 – are baseline values available for the analytes investigated in healthy individuals? For example, Transgelin, 84.1 ng/ml – how elevated is this values with respect to individuals with no renal damage?

The values for transgelin were significantly changed when comparing male v female. Any explanation for this and would this necessitate different thresholds for males/females?

SMM by definition has normal renal function. How do you explain that the TG levels are higher in SMM?

If SMM samples are removed from the analysis of ISS (panel B, Fig. 1), are the results significant?

Treated v untreated comparison. Are all the patient treated using the same therapeutics? Could this group be further refined based on this information?

In the maintenance group TG levels are lower. Is this due to the use of bioogical agents eg Revlimid?

Higher baseline serum transgelin significantly predicted lower eGFR values after the follow-up, independently of baseline eGFR and other covariates. What other covariates were evaluated?

Can the differences between model 1 & 2 be more clearly defined?

Discussion: How is renal damage treated clinically in Myeloma patients currently and has this approach changed recently?

More discussion of the results is needed, only a few lines is based on the results, most of the discussion relates to other researchers findings and does not connect with this manuscript.

Methods: control samples are included in the methods section – is this cohort used in the results displayed?

Human TAGLN ELISA Kit has a minimum detectable dose 0.625-40 ng/ml, and the intra- and interassay precision of 8% and 10%, respectively. Can the authors clarify minimum detectable dose 0.625-40 ng/ml? Also, is the precision data based on this groups data?

Were samples run in duplicate/triplicate?

What quality control criteria were used for the collection of serum samples? More detail is needed here with respect to sample processing.

Overall, the conclusion that TG is a marker of renal damage is not supported by the data. The higher levels in SMM where renal function is normal doesn''t support this. IHC on a cohort who had renal biopsy would help to localise the TG expression. Control and MGUS samples as well as samples from patients with renal impairment due to non-Myelomatous disease eg Diabetes, HTN would be important.

Author Response

Response to Reviewer 1 

The authors thank the Reviewer for a thorough evaluation of the manuscript. We have carefully addressed the comments of all Reviewers. The changes introduced in the text of the manuscript have been marked using red font. Below, we present the detailed answers to the Reviewer’s comments and the description of the modifications introduced upon revision of the manuscript.

Abstract: needs some work, the methods and patient samples used need to be more clearly defined in this section

Response: Upon revision, we have extended ‘materials and methods’ in abstract: “The study included 126 ambulatory patients with multiple myeloma (MM). Serum transgelin-2 concentrations were measured by enzyme-linked immunoassay. We evaluated associations between baseline transgelin and kidney function (serum creatinine, estimated glomerular filtration rate – eGFR, urinary markers of tubular injury: cystatin-C, neutrophil gelatinase associated lipocalin – NGAL monomer, cell cycle arrest biomarkers IGFBP-7 and TIMP-2) and markers of MM burden. Baseline serum transgelin was also evaluated as a predictor of kidney function after a follow-up of 27 months from the start of the study.” (lines 26-33).

Introduction: What was the reason to investigate transgelin in this study? Was this based on a literature review or had this group some previous findings on transgelin that prompted this study?

Response: Transgelin has been linked to kidney fibrosis in several previous studies. Non-reversible kidney insufficiency affects a part of patients with multiple myeloma, which has been associated with adverse prognosis. Also, upregulation of transgelin has been observed in neoplasms. We hypothesized that serum transgelin might predict chronic kidney insufficiency in MM patients.

In the revised manuscript, we have modified the ‘Introduction’ substantially, including the short summary of the previous findings showing the association of transgelin with (1) tumorigenesis and (2) kidney fibrosis and insufficiency. We have clearly specified the aim of the study.

We are grateful for the Reviewer’s remark. We hope that the revised ‘Introduction’ enables better understanding of our assumptions and the purpose of our study.

The following fragment has been added in Introduction, lines 87-104 ” Transgelin-2 (SM22), a cytoskeletal actin-binding protein involved in differentiation of smooth muscle cells, osteoblasts and adipocytes, is present in fibroblasts, some epithelial cells, immune cells (as the only one from transgelin family proteins), bone marrow cells or stem cells18]. Basic function of transgelin-2 is participation in cytoskeleton remodeling via its effect on actin regulation. Moreover, transgelin is involved in bone marrow mesenchymal stem cell (MSC) proliferation and differentiation. Recent studies report dysregulation of SM22 in different malignancies and emphasize its role in cancer development and progression. According to available data, apart from oncogenic role in solid tumors SM22 is also upregulated in leukemia and lymphoma cell lines and takes part in B-cell lymphoma development [19]. Interestingly, transgelin-2 overexpression may be associated with chemotherapy resistance [20].   

    Furthermore, transgelin-2 was found to be a marker of interstitial fibrosis, glomerulosclerosis, and renal damage [21]. Its upregulation depends on the etiology of the disease in various cells (glomerular parietal or visceral, or tubular interstitial cells) and elevated expression was detected in glomerular and in tubulointerstitial injury.

     The aim of our study was to evaluate serum transgelin as a potential marker of renal impairment in patients with MM. We hypothesized that increased serum concentrations of transgelin may be related to irreversible renal insufficiency in this disease.”

Serum creatinine measurement is subject to several caveats to prediction of ongoing renal damage. These caveats need to be explained in detail to put into perspective the need for additional biomarkers of kidney damage.

Response: Thank you for this point. We have added the following text in ‘Introduction’, lines 72-81: “In clinical routine, the laboratory assessment of renal function usually largely relies on serum creatinine measurements used to estimate the glomerular filtration rates. However, routinely available diagnostic measures (e.g. serum creatinine) are subject to several caveats and do not always allow for early and reliable prediction of ongoing renal damage [17]. Serum creatinine concentrations should be interpreted with the awareness of confounding factors such as (1) prerenal azotemia, (2) production rate dependent on individual and clinical characteristics, i.e., age, gender, muscle mass, and medication use, (3) late rise in up to 72 h following injury, (4) large “renal reserve”, and (5) concomitant diseases (e.g., sepsis, liver disease, muscle wasting). Therefore, serum creatinine may not adequately reflect the actual fall in glomerular filtration.”

Results: The patient group studied in this manuscript represent a heterogenous population, how this this affect the analysis and were any inter-group comparison investigated based on response rates?

Response: We described the inter-group comparisons of serum transgelin concentrations in ‘Results’, paragraph 2.2. We did not observe significant differences in serum transgelin between patients with regard to the response to MM treatment (lines 149-157): “In patients who received any MM treatment before entering the study, there was no correlation between serum transgelin and the number of prior treatment lines (R=-0.10; p=0.3). Patients on the maintenance treatment had non-significantly lower serum transgelin than those treatment-naïve (median 77.4 versus 91.0 ng/mL; p=0.051). Treatment regimens included lenalidomide in 25 patients (20%), bortezomib in 21 (17%), thalidomide in 15 (12%), cyclophosphamide in 8 (6%), and melphalan in 3 (2%); steroids were used in 58 (46%) of patients. We did not observe any significant differences in serum transgelin concentrations between patients treated (versus un-treated) with particular drugs (p>0.05 for all comparisons).”

Table 2 – are baseline values available for the analytes investigated in healthy individuals? For example, Transgelin, 84.1 ng/ml – how elevated is this values with respect to individuals with no renal damage?

Response: In Table 2, we have added the information on reference intervals for standard laboratory tests and the results of non-standard laboratory tests (including serum transgelin) obtained in a group of 32 healthy volunteers. Serum transgelin concentrations were significantly higher in MM patients than in controls, also after adjustment for age. In ‘Results’, we have included this information in paragraph 2.1, lines 123-130: “As compared to a control group of 32 healthy volunteers (16 men and 16 women aged 29-74 years), the studied MM patients presented with significantly higher serum concentrations of transgelin and interleukin 6, and higher urinary concentrations of IGFBP-7 and TIMP-2 (Table 2). Although the age range of healthy controls was matched with the age range of patients, the mean age was lower in the control group (50 versus 67 years, p<0.001). However serum transgelin remained higher in patients then in controls after adjustment for the age difference (p=0.034). The sex distribution in the study and control groups was similar (p=0.4).”

Since our control group was small, we have added the following sentence, acknowledging this fact a  limitation of our study: “Moreover, there are no standardized laboratory assays to measure transgelin concentrations. Although we provided the information on serum transgelin in a small group of healthy persons, these data must be considered provisional.” (‘Discussion’, lines 361-363).

The values for transgelin were significantly changed when comparing male vs female. Any explanation for this and would this necessitate different thresholds for males/females?

Response: Although we do not know the reason of this sex-related difference, there are previous reports showing different concentrations or expression of transgelin in males and females. The proteomic analysis of human plasma published by Silliman et al. revealed 14-fold higher transgelin concentrations in men compared to women. Animal studies also reported sex-related differences in the expression of transgelin.

We have added a brief commentary on this issue in ‘Discussion’, lines 276-286:

“In our patients with MM, serum transgelin was significantly higher, as compared to healthy controls. We also observed higher transgelin levels in men with MM, as compared to women. The sex-related difference could not be attributed to differences in MM stage, kidney function or treatment as these were not different between men and women (data not shown). Sex-related differences in circulating transgelin have been, however, reported by others. The proteomic analysis of human plasma published by Silliman at al. [35] revealed 14-fold higher transgelin concentrations in males than in females. Animal studies also reported sex-related differences in the expression of transgelin [36]. Interestingly, in our control group trangelin levels were higher in females. We do not know the reasons for sex-related differences in circulating transgelin concentrations, and further studies are necessary to reveal the underlying causes.”

We have also provided more details on sex-related differences in transgelin concentrations in ‘Results’, paragraph 2.2, lines 142-148 “Median serum transgelin in the whole studied group was 84.1 ng/mL (Table 2). Transgelin concentrations were higher in men (median 96.2 versus 78.8 ng/mL; Figure 1A), in patients with smoldering MM (median 149.2 versus 82.4 ng/mL; p=0.003; Figure 1B) and in treatment-naïve patients (median 145.2 versus 82.5 ng/mL; p=0.014; Figure 3C), however the majority of patients with smoldering MM (5 out of 7) and treatment-naïve patients (6 out of 8) were men. Of interest, serum transgelin was higher in healthy women than in men (control group; median 83.4 vs 60.5 ng/mL, respectively; p=0.011).“

SMM by definition has normal renal function. How do you explain that the TG levels are higher in SMM.

Response: There may be several explanations for higher serum transgelin in SMM, although all remain speculative, considering the lack of previous data.

We explain our point of view in the following fragment that has been added in ‘Discussion’, lines 335-357: “Although smoldering MM (SMM) patientsave normal renal function defined as GFR >60 mL/min/1.73 m2, we found elevated trangelin-2 serum concentrations in this group. This finding may have a pathophysiological explanation. The upregulation of transgelin-2 has been associated with tumorigenesis and cancer development and may vary along with clinical stage and tumor size. Interestingly, several studies revealed higher levels of transgelin-2 in inflammation (i.e. SIRS) and explored SM22 overexpression in the regulation of the NIK transcription and proinflammatory NF-kB signaling pathways as a modulator of vascular inflammation [20,45]. These studies suggest that transgelin may be viewed as an anti-inflammatory marker. Taking into consideration the role of interleukin 6 in MM pathogenesis as a growth and survival factor, inhibiting apoptosis in myeloma cells, this may also explain SM22 role in tumorigenesis. This may support the hypothesis that at the beginning of the disease and tumorigenesis transgelin-2 concentrations are higher. However, a few reports demonstrated that transgelin-2 rather inhibits the motility of cancer cells by suppressing actin polymerization. Moreover, according to available data, only 2% of patients with SMM develop MM. Further, higher concentrations of transgelin in our patients with SMM may possibly be associated with the fact that they had received no treatment. Transgelin levels were also higher in patients who did not received any MM treatment before the study. Moreover, the sex of the patients with SMM may play a role in elevated transgelin-2 concentrations as in the studied group transgelin concentrations were higher in men, and SMM/untreated patients were mostly men. However, since we were not able to identify previous reports on transgelin in patients with SMM, and the number of patients with SMM in our study is very low, we may only speculate on this finding.”

If SMM samples are removed from the analysis of ISS (panel B, Fig. 1), are the results significant?

Response: No, the differences between patients with different ISS stages was non-significant; we mentioned this in ‘Results’, paragraph 2.2: “Transgelin did not differ according to ISS stage (p=0.3).” For clarity, we have modified Figure 1, panel B, to include the message.

Treated vs untreated comparison. Are all the patient treated using the same therapeutics? Could this group be further refined based on this information?

Response: The patients were treated with various schemes, including lenalidomide in 25 patients (20%), bortezomib in 21 (17%), thalidomide in 15 (12%), cyclophosphamide in 8 (6%), and melphalan in 3 (2%); steroids were used in 58 (46%) of patients. Serum concentrations of transgelin did not differ according to drugs used. Although we observed higher concentrations of transgelin in patients who did not received any MM treatment before the study (“treatment-naÑ—ve”), in those previously treated there was no association between serum transgelin and the number of treatment lines received. In the revised manuscript, we have supplemented this information in ‘Results’, paragraph 2.2, lines 149-157 “In patients who received any MM treatment before entering the study, there was no correlation between serum transgelin and the number of prior treatment lines (R=-0.10; p=0.3). Patients on the maintenance treatment had non-significantly lower serum transgelin than those treatment-naïve (median 77.4 versus 91.0 ng/mL; p=0.051). Treatment regimens included lenalidomide in 25 patients (20%), bortezomib in 21 (17%), thalidomide in 15 (12%), cyclophosphamide in 8 (6%), and melphalan in 3 (2%); steroids were used in 58 (46%) of patients. We did not observe any significant differences in serum transgelin concentrations between patients treated versus untreated with particular drugs (p>0.05 for all comparisons).”

In the maintenance group TG levels are lower. Is this due to the use of biological agents e.g. Revlimid?

Response: We are grateful for the Reviewer’s remark. We have supplemented the ‘Results’ with the information on treatment modalities, as specified above.

Moreover, we have briefly discussed the issue (lines 318-334) “In our study, patients were treated with various regimens, most commonly including lenalidomide (in 20% of patients). Serum concentrations of transgelin did not differ according to drugs used. Although we observed higher concentrations of transgelin in patients who did not receive any MM treatment before the study, in those previously treated there was no association between serum transgelin and the number of treatment lines received. Specifically, serum transgelin levels did not differ between those who received lenalidomide and those who did not. Lenalidomide modulates different components of the immune system by interactions with cytokine production through T-cell and NK cell regulation. It is associated with inhibition of pro-inflamatory cytokines interleukin 6 and tumor necrosis factor α (TNF-α). Furthermore, lenalidomide inhibits MM cells and their interactions, leading to apoptosis [44]. In our study, there was no correlation between serum interleukin 6 and transgelin levels in MM patients (R=0.09; p=0.4). Considering that only 25 patients received lenalidomide at the start of our study, we cannot reliably exclude a weak to moderate effect of the drug (or other anti-MM medications) on serum transgelin concentrations. Since transgelin expression has been studied only as an overexpressed molecule in MM transformation to PCL, future studies should reveal the role of transgelin in the development of MM and how it may be affected by MM treatment.”

Higher baseline serum transgelin significantly predicted lower eGFR values after the follow-up, independently of baseline eGFR and other covariates. What other covariates were evaluated?

Response: The variables included in the models were previously listed in Table 3 and Table 3 footnote. In the revised version of the manuscript, the relevant text of ‘Results’ has been modified and now it includes the complete information about covariates included in the model: “…higher baseline serum transgelin significantly predicted lower eGFR values after the follow-up period, independently of baseline eGFR, urinary concentrations of tubular injury markers (NGAL monomer and IGFBP-7), sex, age, prior treatment, treatment response, and observation duration (Table 3).” (lines 197-200).

Can the differences between model 1 & 2 be more clearly defined?

Response: There is only one difference between the models: in Model 1, log-transformed serum transgelin is included as a continuous variable; to the contrary, serum transgelin concentrations have been categorized and transgelin tertiles are included in Model 2. Otherwise, the models are the same, i.e. the covariates and the outcome variable do not differ. In the revised manuscript, we have added this explanation in the title of Table 3, lines 210-212: “In Model 1, serum transgelin was included as a continuous variable after log-transformation, while in Model 2, serum transgelin was included after categorization into tertiles (the tertiles’ boundaries were 71.0 and 110.6 ng/mL).”

Discussion: How is renal damage treated clinically in Myeloma patients currently and has this approach changed recently?

Response: We added the information about general treatment in MM and the breakthrough drug, also improving renal impairment in ‘Discussion’, lines 303-312: “The most popular three-drug regimen used in the MM treatment consists of: proteasome inhibitor: bortezomib, immunomodulatory drug: lenalidomide and dexamethasone. In case of relapse or refractory MM, antibodies targeting myeloma cells (e.g. daratumumab, elotuzumab, isatuximab, belantamab mafodotin), nuclear export inhibitors (selinexor) or histone deacetylase inhibitors (panobinostat) are used. One of the most important game changer drugs in MM was proteasome inhibitor bortezomib, owing to its various anti-myeloma effects including disruption of the cell cycle and induction of apoptosis, alteration of the bone marrow microenvironment and inhibition of nuclear factor kappa B (NFκB). This novel agent improves renal function and should be used especially in the group of patients with lower GFRs [41].”

Moreover, we have briefly commented on the results regarding the modality of treatment (‘Discussion’, lines 318-334): “In our study, patients were treated with various regimens, most commonly including lenalidomide (in 20% of patients). Serum concentrations of transgelin did not differ according to drugs used. Although we observed higher concentrations of transgelin in patients who did not receive any MM treatment before the study, in those previously treated there was no association between serum transgelin and the number of treatment lines received. Specifically, serum transgelin levels did not differ between those who received lenalidomide and those who did not. Lenalidomide modulates different components of the immune system by interactions with cytokine production through T-cell and NK cell regulation. It is associated with inhibition of pro-inflamatory cytokines interleukin 6 and tumor necrosis factor α (TNF-α). Furthermore, lenalidomide inhibits MM cells and their interactions, leading to apoptosis [44]. In our study, there was no correlation between serum interleukin 6 and transgelin levels in MM patients (R=0.09; p=0.4). Considering that only 25 patients received lenalidomide at the start of our study, we cannot reliably exclude a weak to moderate effect of the drug (or other anti-MM medications) on serum transgelin concentrations. Since transgelin expression has been studied only as an overexpressed molecule in MM transformation to PCL, future studies should reveal the role of transgelin in the development of MM and how it may be affected by MM treatment.”

More discussion of the results is needed, only a few lines is based on the results, most of the discussion relates to other researchers findings and does not connect with this manuscript.

Response: Based on the Reviewer’s remarks, we have substantially revised the discussion. The following parts have been included in the revised version:

  1. Trangelin-2 in SMM (lines 335-357),
  2. Transgelin-2 associations with treatment schemes (lines 303-334),
  3. Sex-related differences in transgelin-2 (lines 276-286),
  4. Transgelin-2 correlations with studied markers related to renal impairment and with the decrease in GFR (lines 245-251).

Methods: control samples are included in the methods section – is this cohort used in the results displayed?

Response: The control group included 32 healthy individuals who provided blood and urine samples used to obtain reference results of non-standard laboratory tests. In the previous version of the manuscript, we mistakenly informed that there were 21 control subjects. We are sorry for the mistake. This has been corrected in the revised manuscript (‘Materials and methods’, paragraph 4.1, lines 382-385: “Control group was recruited in order to obtain the reference results of non-standard laboratory tests and included 32 healthy volunteers (16 women, 16 men) aged 29 to 74 years.“). We have added the missing information about the laboratory results in healthy controls in Table 2 and the relevant text of the ‘Results’, lines 123-130: “As compared to a control group of 32 healthy volunteers (16 men and 16 women aged 29-74 years), the studied MM patients presented with significantly higher serum concentrations of transgelin and interleukin 6, and higher urinary concentrations of IGFBP-7 and TIMP-2 (Table 2). Although the age range of healthy controls was matched with the age range of patients, the mean age was lower in the control group (50 versus 67 years, p<0.001). However serum transgelin remained higher in patients then in controls after adjustment for the age difference (p=0.034). The sex distribution in the study and control groups was similar (p=0.4).”).

Human TAGLN ELISA Kit has a minimum detectable dose 0.625-40 ng/ml, and the intra- and interassay precision of 8% and 10%, respectively. Can the authors clarify minimum detectable dose 0.625-40 ng/ml? Also, is the precision data based on this groups data?

Response:  Thank you for pointing out the mistakes. We have cited the assay range (detection range) based on the assay’s standard curve. The intra- and interassay precision (8% and 10%) are declared by the test manufacturer. The description of transgelin assay has been corrected in ‘Materials and methods’, paragraph 4.3, lines 427-430: “Serum transgelin was measured using Human TAGLN (Transgelin) ELISA Kit (Wuhan Fine Biotech Co, Wuhan, China), with the detection range of 0.625-40 ng/mL, and the intra- and interassay precision of 8% and 10%, respectively, as reported by the manufacturer of the assay.”

Were samples run in duplicate/triplicate?

Response:  Due to financial constraints, the samples were not measured in replicates.

What quality control criteria were used for the collection of serum samples? More detail is needed here with respect to sample processing.

Response:  In the revised manuscript, we have enclosed detailed information about sample processing in ‘Materials and methods’, paragraph 4.2, lines 401-406: “Blood was collected based on Good Laboratory Practice (GLP) and Good Clinical Practice (GCP) principles by qualified staff. Blood was collected to closed tubes with clot activator. The samples were mixed and kept in the ambient temperature in vertical position for 30 minutes, then centrifuged at 1300 x g for 15 minutes. After centrifugation, serum was aliquoted and kept at -70°C until analysis. We did not use hemolyzed or lipemic samples.”

Overall, the conclusion that TG is a marker of renal damage is not supported by the data. The higher levels in SMM where renal function is normal doesn''t support this. IHC on a cohort who had renal biopsy would help to localise the TG expression. Control and MGUS samples as well as samples from patients with renal impairment due to non-Myelomatous disease eg Diabetes, HTN would be important.

Response: We have modified the conclusions in the revised manuscript (lines 469-475). We admit that serum transgelin in our patients might be associated with various pathological processes, e.g. malignancy, inflammation or treatment. Therefore, serum transgelin cannot be named a marker of kidney damage, even though transgelin expression has been reported in relation to kidney injury and repair processes. However, our main finding is the association between elevated transgelin and decreasing renal function in multiple myeloma. This is a novel finding and, considering the limitations of our study, it must be confirmed in further studies before drawing definitive conclusions.

Reviewer 2 Report

It is a good ms. It talks about transgelin-2 uesd to  investigate renal damage. The whole text have great layout and I just want to know the follow-up duration of your research. 

other minor comment: p should be Italic and ml should be mL
Am J Hematol . 2017 Mar;92(3):269-278. doi: 10.1002/ajh.24634. Epub 2017 Feb 1.

Author Response

Response to Reviewer 2 

The authors thank the Reviewer for a thorough evaluation of the manuscript. We have carefully addressed the comments of all Reviewers. The changes introduced in the text of the manuscript have been marked using red font. Below, we present the detailed answers to the Reviewer’s comments and the description of the modifications introduced upon revision of the manuscript.

It is a good ms. It talks about transgelin-2 used to investigate renal damage. The whole text have great layout and I just want to know the follow-up duration of your research. 

Response: The follow-up was performed after 27 months from the start of the study. We have added this information in ‘Materials and methods’, paragraph 4.1. lines 382-385: “The follow-up data were collected after 27 months from the start of the study, and included (1) the date and the cause of death, (2) the results of laboratory tests, including serum creatinine and eGFR obtained at the final follow-up visit.”

The actual observation time was shorter than 27 months in many cases due to later recruitment or loss from observation (death, change of treatment center). The median observation time was 21 months, as specified in ‘Results’, paragraph 2.2.

Other minor comment: p should be Italic and ml should be mL.

Response: We have corrected these throughout the manuscript, including figures.

Am J Hematol . 2017 Mar;92(3):269-278. doi: 10.1002/ajh.24634. Epub 2017 Feb 1.

Response: Thank you for this valuable remark. In the revised ‘Discussion’ (lines 312-317), we have added a brief paragraph regarding the relationship between MM treatment and IKAROS protein levels: Moreover, Bolomsky et al. found the association between gene expression levels of several immunomodulatory drug targets in bone marrow mononuclear cells of MM patients and response to the lenalidomide-dexamethasone regimen [42]. Interstingly, high IKAROS protein levels are associated with successful outcome in MM patients [42]. IKAROS was also found among smooth muscle genes in renin cell in the kidney [43].”

Moreover, the ‘Introduction’ and ‘Discussion’ have been substantially revised following the comments of two other Reviewers. In Results, we have added the comparison between MM patients and healthy controls and the comparison between MM patients treated with various drugs. Grammar and language mistakes have been corrected throughout the manuscript.

Reviewer 3 Report

This research investigated the relationship between transgelin-2 and the renal function in patients with multiple myeloma (MM), which found that transgelin-2 may be useful in investigating renal damage in MM patients as a biomarker of injury to tubuloglomerular compartments, which may reflect developing chronic kidney disease. Transgelin-2 is a new biomarker, which makes this research innovative.

Major comments:

  1. The clinical significance of this study is unclear. Based on the results of Table 3, we can find that higher baseline transgelin-2 and lower baseline eGFR were both significantly associated with lower eGFR after follow-up. What is the advantage of transegelin-2 when we take it as a predictor of renal damage, as eGFR is a more common used biomarker in clinical practice.
  2. The author concluded that transgelin-2 may be useful in investigating renal damage in MM patients as a biomarker of injury to tubuloglomerular compartments. However, this study did not include renal pathology information, nor was it to compare the difference between transegelin-2 and other biomarkers of tubular injury, so this conclusion could not be drawn.
  3. Transgelin-2 has no normal value range and is non-normally distributed, which limits its clinical application.
  4. It’s not appropriate to use regression model to look for independent predictors of transgelin-2 and eGFR at the end of follow-up, Because survival time is an issue that has to be considered.
  5. The confounders were not well adjusted in this research.

Minor comments:

Is this study a prospective cohort study? What is the follow-up duration? What is the time interval between two follow-ups?

A control group included 21 healthy volunteers was included in this study, but I did not find any use and results of it. What is the purpose of setting up a control group?

Author Response

Response to Reviewer 3

The authors thank the Reviewer for a thorough evaluation of the manuscript. We have carefully addressed the comments of all Reviewers. The changes introduced in the text of the manuscript have been marked using red font. Below, we present the detailed answers to the Reviewer’s comments and the description of the modifications introduced upon revision of the manuscript.

  1. The clinical significance of this study is unclear. Based on the results of Table 3, we can find that higher baseline transgelin-2 and lower baseline eGFR were both significantly associated with lower eGFR after follow-up. What is the advantage of transegelin-2 when we take it as a predictor of renal damage, as eGFR is a more common used biomarker in clinical practice.

Response: Our findings indicate that baseline serum transgelin and eGFR are independent predictors of final eGFR, which means that when we know both of them, we may predict the final eGFR better than using single predictor. Available literature indicates that transgelin expression in kidneys is associated with injury and fibrosis. Physiologically transgelin is not present in adult’s glomeruli apart from damaged ones. Therefore, transgelin may be sensitive marker of kidney injury, especially in cases of glomerular damage, and its’ higher level may appear earlier or independently than decrease in GFR (e.g. in early stages of nephrotic syndrome where damage of glomeruli is observed with normal GFR).  

  1. The author concluded that transgelin-2 may be useful in investigating renal damage in MM patients as a biomarker of injury to tubuloglomerular compartments. However, this study did not include renal pathology information, nor was it to compare the difference between transegelin-2 and other biomarkers of tubular injury, so this conclusion could not be drawn.

Response: Although we have measured urinary concentrations of several markers of tubular injury, i.e. NGAL monomer, cystatin C, and cell cycle arrest biomarkers TIMP-2 and IGFBP-7, we only found weak significant correlation between serum transgelin and urine cystatin C. Moreover, serum transgelin correlated positively with serum concentrations of FLC lambda (the type of FLC more often associated with renal injury in MM). However, our main finding was the association between baseline serum transgelin and final eGFR. Although serum transgelin cannot be unambiguously named a marker of kidney damage, our results show that its concentrations predict long-term irreversible kidney insufficiency in patients with MM.

We have modified the conclusions to better reflect the findings of the study: “Our study shows for the first time that elevated serum transgelin is negatively associated with glomerular filtration rate in MM and predicts a decrease in renal function over long-term follow-up. However, elevated serum transgelin in MM might be associated with other pathological processes, e.g. malignancy, or inflammation, as well as with MM treatment. Considering the limitations of our study, further work is needed to evaluate transgelin expression in various types of renal involvement in MM. Still, our findings support previous reports that link transgelin to kidney fibrosis” (lines 469-475).

  1. Transgelin-2 has no normal value range and is non-normally distributed, which limits its clinical application.

Response: In Table 2, we have added the information on reference intervals for standard laboratory tests and the results of non-standard laboratory tests (including serum transgelin) obtained in a group of 32 healthy volunteers. Serum transgelin concentrations were significantly higher in MM patients than in controls, also after adjustment for age. In ‘Results’, we have included this information in paragraph 2.1, lines 123-130.

Since our control group was small, we have added the following sentence, acknowledging this fact a  limitation of our study: “Moreover, there are no standardized laboratory assays to measure transgelin concentrations. Although we provided the information on serum transgelin in a small group of healthy persons, these data must be considered provisional.” (‘Discussion’, lines 361-363).

  1. It’s not appropriate to use regression model to look for independent predictors of transgelin-2 and eGFR at the end of follow-up, because survival time is an issue that has to be considered.

Response: The use of survival analysis would be appropriate if we looked for the time to eGFR decrease. However, we were interested in long-term renal insufficiency in the studied patients. Therefore, we collected the data on final eGFR at the end of follow-up. The Cox regression does not fit well to such data (it cannot model a continuous outcome variable without a censoring variable). The follow-up time was included as a confounder in the analysis presented in Table 3.

                In the revised manuscript, we have included the data on mortality (‘Results’, paragraph 2.3, lines 178-180): “Twenty-three patients (18%) died during the study period, including 12 due to MM, 7 due to infection, and 3 due to undefined reason.” and (lines 206-208): “By using simple Cox regression, we did not observe any association between serum transgelin and overall or MM-specific survival, neither in the entire study group, nor after exclusion of patients with smoldering MM (p>0.5 in all cases).”

  1. The confounders were not well adjusted in this research.

Response: The multiple regression models presented in Table 3 (showing the association between serum transgelin and final eGFR) were adjusted for other variables significantly associated with final eGFR (i.e. baseline eGFR, tubular markers: urinary NGAL and IGFBP-7, age) and the clinically relevant variables (sex, prior treatment, response to treatment, and the length of observation). A part of this information has been presented in the footnote to Table 3 in the previous version of the manuscript.

In the revised version, we have included this information in the text of ‘Results’, paragraph 2.3, lines 197-205: ”Moreover, higher baseline serum transgelin significantly predicted lower eGFR values after the follow-up period, independently of baseline eGFR, urinary concentrations of tubular injury markers (NGAL monomer and IGFBP-7), sex, age, prior treatment, treatment response, and observation duration (Table 3). Moreover, transgelin values in the upper tertile (i.e. above 110.6 ng/mL) were independently associated with lower eGFR at the end of observation (Figure 2; Table 3). Although baseline uNGAL (R=-0.31; p<0.001) and uIGFBP-7 (R=-0.35; p<0.001) were also significantly associated with final eGFR, these associations did not prove to be independent of other predictors, including baseline eGFR (Table 3).”

We have also added the details regarding the construction of the models in ‘Materials and methods’, paragraph 4.4 (Statistical analysis).

  1. Is this study a prospective cohort study? What is the follow-up duration? What is the time interval between two follow-ups?

Response: We have added a statement in ‘Materials and methods’, paragraph 4.1: “This was a prospective observational study”.

Patients were recruited during ambulatory control visits at the Departments of Hematology of the University Hospital in Kraków, Poland, as stated in ‘Material and methods’, paragraph 4.1. The follow-up data were collected after 27 months from the start of the study. We have added this information in ‘Materials and methods’, paragraph 4.1. lines 382-385: “The follow-up data were collected after 27 months from the start of the study, and included (1) the date and the cause of death, (2) the results of laboratory tests, including serum creatinine and eGFR obtained at the final follow-up visit.”

The actual observation time was shorter than 27 months in many cases due to later recruitment or loss from observation (death, change of treatment center). The median observation time was 21 months, as specified in ‘Results’, paragraph 2.2.

  1. A control group included 21 healthy volunteers was included in this study, but I did not find any use and results of it. What is the purpose of setting up a control group?

Response: The control group included 32 healthy individuals who provided blood and urine samples used to obtain reference results of non-standard laboratory tests. In the previous version of the manuscript, we mistakenly informed that there were 21 control subjects. We are sorry for the mistake. This has been corrected in the revised manuscript (‘Materials and methods’, paragraph 4.1, lines 386-388: ”Control group was recruited in order to obtain the reference results of non-standard laboratory tests and included 32 healthy volunteers (16 women, 16 men) aged 29 to 74 years.”).

We have added the missing information about the laboratory results in healthy controls in Table 2 and the relevant text of the ‘Results’, lines 123-130: “As compared to a control group of 32 healthy volunteers (16 men and 16 women aged 29-74 years), the studied MM patients presented with significantly higher serum concentrations of transgelin and interleukin 6, and higher urinary concentrations of IGFBP-7 and TIMP-2 (Table 2). Although the age range of healthy controls was matched with the age range of patients, the mean age was lower in the control group (50 versus 67 years, p<0.001). However serum transgelin remained higher in patients then in controls after adjustment for the age difference (p=0.034). The sex distribution in the study and control groups was similar (p=0.4)”.

Round 2

Reviewer 3 Report

The author has addressed all my comments and revised the manuscript. I agree to accept this manuscript in its present form.